# Enhanced Cd-Accumulation in *Typha latifolia* by Interaction with *Pseudomonas rhodesiae* GRC140 under Axenic Hydroponic Conditions

**DOI:** 10.3390/plants11111447

**Published:** 2022-05-29

**Authors:** Gisela Adelina Rolón-Cárdenas, Joana Guadalupe Martínez-Martínez, Jackeline Lizzeta Arvizu-Gómez, Ruth Elena Soria-Guerra, Ma. Catalina Alfaro-De la Torre, Fulgencio Alatorre-Cobos, Jesús Rubio-Santiago, Regina de Montserrat González-Balderas, Candy Carranza-Álvarez, José Roberto Macías-Pérez, Liseth Rubí Aldaba-Muruato, Alejandro Hernández-Morales

**Affiliations:** 1Facultad de Ciencias Químicas, Universidad Autónoma de San Luis Potosí, San Luis Potosí 78210, Mexico; gisela.rolon@uaslp.mx (G.A.R.-C.); ruth.soria@uaslp.mx (R.E.S.-G.); alfaroca@uaslp.mx (M.C.A.-D.l.T.); jesusrub92@gmail.com (J.R.-S.); regina.gonzalez@uaslp.mx (R.d.M.G.-B.); candy.carranza@uaslp.mx (C.C.-Á.); 2Facultad de Estudios Profesionales Zona Huasteca, Universidad Autónoma de San Luis Potosí, Ciudad Valles, San Luis Potosí 79060, Mexico; joanaguadalupem@gmail.com (J.G.M.-M.); roberto.macias@uaslp.mx (J.R.M.-P.); liseth.aldaba@uaslp.mx (L.R.A.-M.); 3Secretaría de Investigación y Posgrado, Centro Nayarita de Innovación y Transferencia de Tecnología (CENITT), Universidad Autónoma de Nayarit, Tepic, Nayarit 63173, Mexico; 4Colegio de Postgraduados Campus Campeche, Campeche 24450, Mexico; fulgencio@colpos.mx

**Keywords:** *Typha*, axenic hydroponic system, PGPB, auxin, phytoremediation

## Abstract

The *Typha* genus comprises plant species extensively studied for phytoremediation processes. Recently, *Pseudomonas rhodesiae* GRC140, an IAA-producing bacterium, was isolated from *Typha latifolia* roots. This bacterium stimulates the emergence of lateral roots of *Arabidopsis thaliana* in the presence and absence of cadmium. However, the bacterial influence on cadmium accumulation by the plant has not been determined. Moreover, the *P. rhodesiae* GRC140 effect in Cd phytoextraction by *T. latifolia* remains poorly understood. In this work, an axenic hydroponic culture of *T. latifolia* was established. The plants were used to evaluate the effects of cadmium stress in axenic plants and determine the effects of *P. rhodesiae* GRC140 and exogenous indole acetic acid (IAA) on Cd tolerance and Cd uptake by *T. latifolia*. Biomass production, total chlorophyll content, root electrolyte leakage, catalase activity, total glutathione, and Cd content were determined. The results showed that Cd reduces shoot biomass and increases total glutathione and Cd content in a dose-dependent manner in root tissues. Furthermore, *P. rhodesiae* GRC140 increased Cd translocation to the shoots, while IAA increased the Cd accumulation in plant roots, indicating that both treatments increase Cd removal by *T. latifolia* plants. These results indicate that axenic plants in hydroponic systems are adequate to evaluate the Cd effects in plants and suggest that *T. latifolia* phytoextraction abilities could be improved by *P. rhodesiae* GRC140 and exogenous IAA application.

## 1. Introduction 

Environmental pollution by heavy metals has increased due to human activities and industrial processes [1]. Cadmium (Cd) is one of the most toxic heavy metals to humans, even at low concentrations [2]. Therefore, different methods such as encapsulation, electrokinetic extraction, stabilization and soil washing, have been developed to reduce Cd concentrations in the environment and thus avoid its deleterious effects on human health and ecosystems [3]. Phytoremediation is an eco-friendly strategy that uses plants and microorganisms to eliminate heavy metals from the contaminated environment [4].

The *Typha* genus comprises plant species used in phytoremediation, *T. domingensis*, *T. angustifolia*, and *T. latifolia* being the most utilized species for such purposes [5]. They can grow in polluted sites that contain mining, domestic and industrial wastewater [6]. So, *Typha* species have been used in tertiary wastewater treatment due to their ability to remove and accumulate Cd, arsenic (As), and lead (Pb), mostly in their root systems [7].

Recently, bacterial-assisted phytoremediation has attracted much attention due to the potential of bacteria to increase plant growth, reduce the adverse effects of environmental stress, and enhance heavy metal phytoaccumulation by modulating endogenous phytohormone production [8,9]. Auxins are important phytohormones that regulate the development and stress response of plants [10]. Exogenous auxin application has been shown to alleviate Cd toxicity and increase Cd accumulation by plants. In *A. thaliana* plants, the application of exogenous naphthaleneacetic acid (NAA) increased Cd tolerance and Cd retention in the roots [11]. Moreover, auxin-producing bacteria tolerant to heavy metals are an excellent alternative to supply or restore auxin production in plants, thus improving the phytoremediation strategies [9].

*Pseudomonas rhodesiae* GRC140 is a Cd-tolerant endophytic bacterium isolated from the roots of *T. latifolia* collected in a Cd contaminated site. This bacterium produces auxins, synthesizes siderophores, solubilizes phosphates, exerts 1-aminocyclopropane-1-carboxylic acid (ACC) deaminase activity, and stimulates lateral root emergence in *A. thaliana* seedlings exposed and non-exposed to 2.5 mg/L of Cd [12]. However, the effects of *P. rhodesiae* GRC140 and its main auxin (IAA) on Cd accumulation by *T. latifolia* remain poorly understood. 

Recently, hydroponic systems have been developed to analyze plant response to nutrient availability and plant–microbe interactions [13,14]. Axenic hydroponic systems use liquid nutrient solutions instead of the soil and provide a more precise control of the growth media composition [15]. These systems have been used to dissect the complex interactions between plant, soil, and microorganisms and determine the influence of microorganisms and their metabolites on plant physiology and heavy metal bioremediation [16].

Therefore, an axenic hydroponic culture of *T. latifolia* was established. These plants were used to (1) evaluate the effects of different Cd concentrations on plant growth; (2) quantify Cd uptake by axenic plants; (3) determine the effect of *P. rhodesiae* GRC140 and IAA on plant growth and Cd uptake.

## 2. Results and Discussion

### 2.1. Plant Authentication

The samples used in this study were obtained from *Typha* plants previously identified as *T. latifolia* by Rodríguez-Hernández, et al. [17]. Plant identity was established by amplifying and sequencing the psbA-trnH intergenic spacer, a marker in the plant chloroplast genome used in molecular taxonomy and plant phylogenetic relationship [18]. A sequence of 646 base pairs (bp) yielded by Sanger sequencing (GeneBank access number MZ905385) was searched in the database using the BLAST algorithm. The sequence showed 99.73% similarity with psbA-trnH intergenic sequences of *T. latifolia* available in the National Center for Biotechnology Information (NCBI) database. These results confirmed the plant identity as *T. latifolia.*

### 2.2. T. latifolia Culture under Axenic Hydroponic Conditions

Axenic *T. latifolia* plants have been previously obtained by in vitro micropropagation from mature inflorescences [19,20], leaf sections [20], and germinated seedlings [21,22]. In this work, axenic *T. latifolia* plants were obtained by seed germination in 0.2× Murashige and Skoog (MS) agar. Under in vitro conditions, the *T. latifolia* seeds showed 60% germination (Appendix A), and ten days after sowing showed more vigorous growth of shoots in the 0.2× MS agar (Appendix A). The *T. latifolia* seedlings were transferred to a sterile hydroponic system that contained 0.2× MS liquid medium supplemented with glucose as the carbon source. The *T. latifolia* plants developed shoots and roots in the hydroponic system after 10 days of growth, while after 60 days of growth *T. latifolia* plants showed good shoot and root development without visual stress signs (Figure 1a). The initial hydroponic solution was maintained during plant development for 60 days. After incubation, no bacterial or fungal contamination was detected in the hydroponic solution, indicating the aseptic conditions of the system (Figure 1b). In contrast with previous studies to obtain axenic *T. latifolia* plants [20,21,23], our hydroponic system did not use phytohormones for plant differentiation, reducing the time to obtain axenic cultures, and allowing us to carry out research with *T. latifolia* plants under nutrition-controlled conditions [24].

### 2.3. Chemical Modeling of Cd Bioavailability in Hydroponics without Glucose

Heavy metal toxicity in plants is closely related to their bioavailability in the hydroponic solution and is influenced by pH, organic matter, ligands, and heavy metal chemical speciation [25]. Complete MS medium has been previously used to evaluate Pb, chromium (Cr), and manganese (Mn) removal by in vitro root culture of *T. latifolia* [26]. In this work, a diluted MS medium (0.2× strength) without glucose was used to ensure Cd bioavailability in the liquid medium. The prediction analysis by chemical modeling showed that all the CdCl_2_ was available in different speciation forms, and no Cd precipitation occurred (Table 1). According to our analysis, the ionic form Cd^2+^ was the predominant species in the 0.2× MS medium at pH 5.7, which was available at 66, 75, and 79% when supplemented with 10, 20, and 40 mg/L of Cd (provided as CdCl_2_), respectively (Table 1). These levels of available Cd are lower than those observed in the Hoagland solution (pH 6.0) where Cd^2+^ availability remains between 80–81% when the solution contains 5–100 µM CdCl_2_ (0.9–18.3 mg/L) [27].

### 2.4. Cadmium Toxicity Effects on T. latifolia in Hydroponic Culture

Sixty-day-old *T. latifolia* plants were exposed to 10, 20, and 40 mg/L of Cd under aseptic hydroponic conditions for ten days. Although a reduction in shoot biomass in a Cd dose-dependent manner was observed after Cd exposure (Figure 2b,c), the Cd concentrations tested did not significantly affect either the phenotype or the chlorophyll content of *T. latifolia* plants (Figure 2a,d). A reduction in dry shoot weight could be due to a significant decrease in the surface area of the leaves caused by the Cd, similar to that observed in *Triticum aestivum* leaves [28].

Electrolyte leakage is a parameter that measures damage to the cell membrane during plant stress [29]. In the present study, a significant increase in root electrolyte leakage was found in plants exposed to 20 and 40 mg/L Cd (Figure 2e). Previous studies have reported that electrolyte leakage is induced by the increase of reactive oxygen species (ROS) [30]. For instance, in *T. angustifolia*, Cd exposure increases superoxide (O_2_^−^) production in a concentration-dependent manner [31]. It is known that ROS activates membrane ion channels that release intracellular potassium (K^+^) ions, stimulating intracellular hydrolytic enzymes that lead to cell death [30,32]. Although ROS levels were not determined in this study, the results suggest that the root electrolyte leakage in *T. latifolia* is induced by the ROS generated in the plants exposed at 20 and 40 mg/L of Cd.

Enzymatic and non-enzymatic antioxidant systems are induced in the plant cells to maintain ROS balance [33]. This study evaluated antioxidant defense by quantifying catalase (CAT) activity and total glutathione (GSH) content. Catalase is one of the main antioxidant enzymes that reduce ROS damage by decreasing intracellular hydrogen peroxide (H_2_O_2_) levels [33]. We observed that CAT activity increased two-fold in *T. latifolia* roots exposed to 10 mg/L of Cd, but no significant changes were detected in shoots and roots exposed to 20 or 40 mg/L of Cd (Figure 3a). Similar results were observed in *Sassafras tzumu* and *Triticum aestivum* plants exposed to Cd [34,35]. The increase in CAT activity at low Cd levels (10 mg/L) may be attributed to increased ROS production induced by Cd oxidative stress, similar to that observed in *T. aestivum* [35]. On the other hand, the low CAT activity at high Cd concentrations could result from Cd ions interacting with amino acids of the CAT active center, preventing CAT binding to H_2_O_2_ by steric hindrance conditions [36]. These results showed that low Cd levels induce CAT activity, while high Cd levels inhibit CAT activity in *T. latifolia* plants.

Glutathione (GSH) is an antioxidant compound that regulates H_2_O_2_ levels by the ascorbate–glutathione pathway in plants [37]. GSH is also a precursor of phytochelatins, which form complexes with heavy metals and transport them to vacuoles wherein they are sequestered and fixed as nontoxic elements [37]. We observed that Cd stress induces the synthesis of total GSH in *T. latifolia* only in root tissues. In plants exposed to 10, 20, and 40 mg/L of Cd, total GSH content in roots increased 21-, 24-, and 36-fold compared to non-exposed plants (Figure 3b). These results are in line with those previously found in *T. angustifolia*, where Cd exposure increased GSH and phytochelatin synthesis to decrease Cd phytotoxicity [1,38]. Our results are consistent with the hypothesis that GSH accumulation could be associated with Cd tolerance in *T. latifolia* plants.

### 2.5. Cd Accumulation and Removal by T. latifolia Plants in the Hydroponic System

Cd accumulation in *T. latifolia* roots and shoots was determined after ten days of Cd exposure under hydroponic conditions. In both organs, Cd levels increased in a dose-dependent manner. The Cd content in roots ranged from 2970 ± 65 to 3520 ± 105 mg/Kg of dry weight (DW), while in the shoots, it was from 406 ± 59 to 739 ± 24 mg/Kg DW (Table 2). The results showed that Cd concentration in *T. latifolia* roots was from 4.7 to 7.3-fold higher than in shoots, suggesting a low efficiency of Cd translocation. Cd levels in *T. latifolia* roots were higher than those reported in *T. latifolia* (1230–2339 mg/Kg) [39], *T. angustifolia* (1213–2977 mg/Kg) [40], and *T. orientalis* (300–1100 mg/Kg) [41] where other culture conditions were used.

Bioconcentration (BCF) and translocation (TF) factors are two indicators used to evaluate the plant’s ability to accumulate heavy metals [42]. BCF refers to the metal absorbed by the plant in the roots from the substrate, while TF indicates the ability of the plant to transport the metal from roots to shoots [43]. In this study, *T. latifolia* BCF ranged from 126 ± 16 to 509 ± 96, while TF ranged from 0.18 ± 0.04 to 0.24 ± 0.01 for Cd (Table 2). These results indicated that *T. latifolia* plants took up Cd from the hydroponic medium accumulating in their roots and did not efficiently translocate it to the shoots. These results make sense with total chlorophyll and total GSH content detected in shoots that seemed not significantly changed by Cd exposure. Similar values of TF and BCF have been reported in *T. angustifolia* [31,38,40] and *T. orientalis* [41]. The low Cd-translocation efficiency in this study could be explained by the findings of Xu, et al. [38] in *T. angustifolia* where the Casparian band present in plant roots prevents Cd mobilization towards vascular tissues, reducing Cd accumulation in shoots.

Moreover, the level of Cd in the hydroponic solution was determined. Overall, *T. latifolia* plants removed 37, 43, and 22% of the Cd from the growth media when exposed to 10, 20, and 40 mg/L of Cd, respectively (Table 2). The results showed that the highest Cd removal efficiency occurred at lower concentrations (10 and 20 mg/L of Cd). Similarly, Alonso-Castro, et al. [39] reported that young *T. latifolia* plants removed 41 and 38% of Cd from aqueous solution when exposed to solutions containing 5.0 and 7.5 mg/L of Cd, respectively.

### 2.6. Effect of P. rhodesiae GRC140 and IAA on T. latifolia Exposed to Cd

Different studies have shown that heavy metal-tolerant plant growth-promoting rhizobacteria (PGPR) improve the phytoremediation processes [44], reduce heavy metal stress in plants, and contribute to plant growth by producing bacterial auxins [9].

Here, we also tested the potential effects of auxin-producing *P. rhodesiae* GRC40 and exogenous IAA on Cd removal capacity by *Typha* plants. For this, the bacterial density in the hydroponic medium and bacterial colonization of *T. latifolia* roots were determined. At 16 days post inoculation (dpi), the number of *P. rhodesiae* GRC140 in the hydroponic medium was log_10_ 6.6 ± 0.2, while in *T. latifolia* roots, it was log_10_ 3.9 ± 0.02. No bacterial morphotypes other than *P. rhodesiae* GRC140 were observed. These results showed that *P. rhodesiae* GRC140 remains viable in the hydroponic systems and can colonize *T. latifolia* roots in the presence of 40 mg/L of Cd. Similar results were reported by Wu, et al. [8], who observed that *Pseudomonas fluorescens* colonizes the roots of *Sedum alfredii* plants exposed to 25 µM (5.9 mg/L) of Cd(NO_3_)_2_.

After Cd exposure, *T. latifolia* plants inoculated with *P. rhodesiae* GRC140 (Cd + GRC140 treatment) showed a phenotype similar to plants of Cd + IAA and Cd treatments (Figure 4a), and we did not detect significant changes in fresh weight, dry weight, and electrolyte leakage (Figure 4b,c,e), while exogenous IAA significantly increased only the total chlorophyll content in *T. latifolia* plants exposed to Cd (Figure 4d). These results agree with Zhou, et al. [45], who observed that exogenous IAA application increased total chlorophyll content in *Cinnamomum camphora* treated with 30 mg/Kg of Cd.

Auxin-producing bacteria have been reported to protect plants against oxidative stress caused by Cd, increasing the activity of antioxidant enzymes and the synthesis of antioxidant molecules [9]. Therefore, we determined the effect of *P. rhodesiae* GRC140 and exogenous IAA on CAT activity and GSH content of *T. latifolia* plants exposed to Cd. Plants inoculated with *P. rhodesiae* GRC140 did not have differences in the CAT activity and total glutathione content in the roots and shoots of *T. latifolia* with respect to control plants (Cd) (Figure 5a,b). Likewise, similar results were observed in *T. latifolia* plants treated with IAA (Figure 5a,b). In our experimental conditions, exogenous IAA and *P. rhodesiae* GRC140 did not increase the antioxidant defense system in *T. latifolia* exposed to Cd.

### 2.7. Effect of P. rhodesiae GRC140 and IAA on Cd Uptake

The Cd content in *T. latifolia* plants treated with *P. rhodesiae* GRC140 or exogenous IAA was analyzed by atomic absorption spectrophotometry (AAS). As shown in Figure 6, *P. rhodesiae* GRC140 increased two-fold the Cd content, and Cd accumulation in shoots compared to the control treatment (Cd) but had no significant effect on the Cd content or Cd accumulation in roots (Figure 6a,b). Furthermore, *P. rhodesiae* GRC140 inoculation significantly increased the TF value (Figure 6c). These data indicated that *P. rhodesiae* GRC140 increased Cd translocation in *T. latifolia* plants. A similar effect has been observed in *Miscanthus floridulus* Lab and *S. alfredii* plants inoculated with *Lelliottia jeotgali* and *Sphingomonas*, respectively [46,47]. An increase in Cd translocation by *P. rhodesiae* GRC140 may be attributed to an increase in the gene expression of Cd transporters similar to that observed in *S. alfredii* [47]. In *S. alfredii*, *Sphingomonas* upregulated NRAMP1 and IRT1 genes encoding metal transporters involved in Cd translocation in *Oryza sativa* [48,49]. However, further research is necessary to understand how *P. rhodesiae* GRC140 may enhance Cd translocation from roots to shoots in *T. latifolia* plants.

On the other hand, exogenous IAA application increased the content and the accumulation of Cd in *T. latifolia* roots but did not affect the Cd translocation factor, suggesting that IAA application increases Cd immobilization in *T. latifolia* plant roots (Figure 6a–c). This effect may be due to IAA increasing the synthesis of cell wall components upon which Cd fixation occurs [11]. Our results are similar to those observed in *A. thaliana* [11] and *Solanum nigrum* [50].

After ten days of incubation, the level of Cd in the hydroponic solution significantly decreased in all treatments (*p* < 0.05). In *T. latifolia* plants, the treatments with IAA or *P. rhodesiae* GRC140 increased by 22 and 8%, the percentage of Cd removal, respectively (Figure 6d). Similar to our results, the inoculation of Cd-resistant microorganisms increased in a 10% removal of Cd^2+^ by *Phragmites australis* in constructed wetlands [51]. Overall, Cd removal from axenic hydroponic systems is consistent with Cd accumulation by *T. latifolia* plants treated with IAA and *P. rhodesiae* GRC140.

## 3. Materials and Methods

### 3.1. Plant Material Sampling and Authentication

Inflorescences and leaves of *Typha* plants were collected from an artificial wetland at the Faculty of Chemical Sciences, UASLP (San Luis Potosí, SLP, Mexico). Molecular authentication of plants was carried out as follows. DNA was obtained from 100 mg of leaves using the CTAB method previously reported [52]. PCR reaction was performed using SuperMix High Fidelity according to the manufacturer’s instructions (Invitrogen, Waltham, MA, USA). The primers psbA 3F (5′GTTATGCATGAACGTAATGCTC3′) and trnH 3R (5′CGCGCATGGTGGATTCACAATCC3′) were used to amplify the psbA-trnH intergenic spacer region located in the plant chloroplast genome [18]. PCR conditions were 95 °C for 10 min; 35 cycles of 95 °C, 30 s; 50 °C, 30 s; 72 °C for 1 min, and a final extension at 72 °C for 5 min. PCR product was purified and sequenced at Biotechnology Institute, UNAM (Cuernavaca, Morelos, Mexico). The sequence obtained was aligned at GeneBank to determine similarity in the NCBI database.

### 3.2. Establishment and Cultivation of Axenic T. latifolia Hydroponic Culture

Mature seeds of *T. latifolia* were surface sterilized with 50% sodium hypochlorite solution and 0.02% Triton X-100 (Sigma-Aldrich, St. Louis, MO, USA) for 15 min, and rinsed six times with sterile distilled water [53]. Then, seeds were germinated on culture flasks with 25 mL MS agar containing Murashige and Skoog (MS; Sigma-Aldrich, St. Louis, MO, USA) salts at 0.2× strength, 3.5 mM 4-Morpholineethanesulfonic acid (MES; pH 5.7, Sigma-Aldrich, St. Louis, MO, USA), 1.0% (*w*/*v*) glucose, and 0.8% (*w*/*v*) Phytagar (Sigma-Aldrich, St. Louis, MO, USA). Culture flasks were incubated at 28 °C under fluorescent light and a photoperiod of 16/8 h light/darkness. Germinated seeds were aseptically transferred to axenic hydroponic systems (three seeds per system) containing 150 mL of 0.2× MS liquid medium, buffer MES (3.5 mM, pH 5.7), and 1.0% (*w*/*v*) glucose [13]. Culture systems were incubated at 28 °C under fluorescent white light with a 16 h light/8 h dark photoperiod for 60 days. Axenic condition of the culture systems was evaluated by plating 20 µL aliquot of the plant-growth medium on LB agar. The LB plates were incubated at 30 °C for five days and then analyzed for microorganism growth.

### 3.3. Cadmium Plant Treatments and Chemical Modeling of Cd Bioavailability

Sixty-day-old *T. latifolia* plants, previously grown in hydroponics, were transferred to a hydroponic system with 0.2× MS medium without glucose for six days for acclimatization. After incubation, the systems (*n* = 3, 12 total systems) were transferred in 0.2× MS liquid medium without glucose at pH 5.7 supplemented with 0, 10, 20, and 40 mg/L of Cd (CdCl_2_) (Fermont, Monterrey, Mexico). Culture systems were incubated at 28 °C for ten days under fluorescent white light with a 16 h light/8 h dark photoperiod. After Cd exposure, *T. latifolia* plants were harvested and fresh and dry weight, total chlorophyll content, root electrolyte leakage, catalase activity, and total glutathione were determined. The total Cd content of roots, shoots, and the hydroponic medium was also determined. Cadmium speciation in the 0.2× MS liquid medium was predicted using the software Visual MINTEQ (Version 3.1, KTH, Stockholm, Sweden). Cd speciation was determined at pH 5.7, introducing into the Software data of ionic concentration of each component of MS basal salts (Catalog M5524; Sigma-Aldrich, St. Louis, MO, USA) and MES buffer. The prediction was made at 10, 20, and 40 mg/L of Cd (CdCl_2_).

### 3.4. P. rhodesiae GRC140 and IAA Treatments in Plants Exposed to Cd

*P. rhodesiae* GRC140 was previously isolated from *T. latifolia* roots by Rolón-Cárdenas, et al. [12]. The bacterial strain was maintained at 4 °C in Luria Bertani (LB) agar plates. For the plant–bacteria interaction, *P. rhodesiae* GRC140 was grown in LB broth and incubated at 30 °C overnight. After incubation, bacterial cells were harvested by centrifugation at 2450 g for 15 min. The bacterial cells were resuspended in 0.01 M MgSO_4_ solution and adjusted to OD_600_ = 1.0 (approximately 10^9^ CFU/mL).

Sixty-day-old *T. latifolia* plants (*n* = 3, 9 total systems) were transferred to 0.2× MS medium without glucose. The hydroponic systems were inoculated with 1.5 mL of 0.01 M MgSO_4_ solution (control), 1.5 mL of *P. rhodesiae* GRC140 suspension in 0.01 M MgSO_4_ and supplemented with indole-3-acetic acid (IAA) at a final concentration of 0.1 µM. Six days after incubation, the medium of all systems was replaced with 0.2× MS medium supplemented with a final concentration of 40 mg/L of Cd (CdCl_2_). The systems were incubated in a plant growth chamber for 10 days in the conditions described above.

After Cd exposure, *T. latifolia* plants were harvested, and fresh and dry weight, total chlorophyll content, root electrolyte leakage, catalase activity, and total glutathione were determined. The total Cd content of roots, shoots, and hydroponic medium, was also determined as described below.

### 3.5. Chlorophyll Quantification

Chlorophyll (Chl) content was determined in the leaves of control and treated plants. Chl extraction of *T. latifolia* plants was carried out according to Lichtenthaler and Buschmann [54]. Fresh leaves (100 mg) were homogenized with 5 mL of acetone, and the extract was kept in darkness for 24 h at 4 °C. The extract absorbance was measured using a UV–Vis spectrophotometer (Evolution 221, Thermo Scientific, Waltham, MA, USA) at 644.8 nm (*A*_644.8_) and 661.6 nm (*A*_661.6_). Chl *a* and Chl *b* were calculated according to Equations (1) and (2), respectively [55].
(1)$$\mathrm{Chl}\;a=11.24\;A_{661.6}-2.040\;A_{644.8}$$
(2)$$\mathrm{Chl}\;b=20.12\;A_{644.8}-4.19\;A_{661.6}$$

Total Chl content was defined as the sum of Chl *a* + Chl *b*. Data were normalized and expressed as mg/g of fresh weight (FW).

### 3.6. Electrolyte Leakage

The electrolyte leakage was determined in the roots of controls and treated plants. The roots were washed with 0.1 M ethylenediaminetetraacetic acid (EDTA) for 15 min to remove any the surface-adsorbed metal and rinsed in deionized water to remove surface ions. Root electrolyte leakage was determined using the electrical conductivity method according to Umnajkitikorn, et al. [56]. The root samples (100 mg) were immersed in 10 mL of deionized water at 25 °C for 1 h. The initial electrical conductivity (ECi) of samples was determined using a conductivity meter (PC2700 Oakton, Cole-Parmer, Vernon Hills, IL, USA). Then, samples were incubated at 20 °C for 24 h, and the final electrical conductivity (ECf) was measured. To determine the total electrical conductivity (ECt), the samples were heated at 120 °C in an autoclave for 15 min. Samples were cooled at 25 °C, and electrical conductivity was determined. Root electrolyte leakage was calculated with the formula:(3)$$\%\;\mathrm{EL}=\left(\frac{\mathrm{ECf}-\mathrm{ECi}}{\mathrm{ECt}-\mathrm{ECi}}\right)\times 100$$
where ECi: initial electrical conductivity, ECf: final electrical conductivity, ECt: total electrical conductivity.

### 3.7. Catalase Activity and Total Glutathione Content

Catalase (CAT) activity and glutathione (GSH) content were determined in the shoots and roots of control and treated plants. Fresh plant tissue (250 mg) was powdered in liquid nitrogen and homogenized with ice-cold 500 uL of 100 mM potassium phosphate buffer pH 7.5 containing 5 mM EDTA (KPE buffer). Samples were centrifuged at 16,000× *g* for 5 min, and the supernatants were collected and used for catalase activity, glutathione total, and protein content.

Total protein was quantified using a standard curve of bovine serum albumin (BSA, Sigma-Aldrich, St. Louis, MO, USA) and Bradford’s reagent (Sigma-Aldrich, St. Louis, MO, USA). Samples were adjusted at 50 µg/mL of total protein, and the catalase activity was determined according to Aebi [57]. A 25 µL aliquot of protein extract was added to the reaction mixture containing 50 mM phosphate buffer (pH 7.0) and 10 mM H_2_O_2_. Decomposition of H_2_O_2_ was monitored by UV–Vis spectroscopy at 240 nm. Enzymatic activity was calculated using the molar extinction coefficient of H_2_O_2_ (39.4 M^−1^ cm^−1^). One CAT unit was defined as the amount of enzyme required to decompose 1 µmol of H_2_O_2_ per minute [58]. CAT activity was normalized with protein content and expressed as U/mg protein.

The total glutathione content was determined using the enzymatic recycling method reported by Rahman, et al. [59]. The protein extract was mixed with sulfosalicylic acid (10%) and Triton X-100 (0.1%) prepared in a KPE buffer. A 100 µL aliquot of protein extract was added to the reaction mixture of 0.05 mM NADPH, 50 U/mL of glutathione reductase (GR, Sigma-Aldrich, St. Louis, MO, USA), and 0.1 mM DTNB, in 100 mM KPE buffer. The reaction mixture was incubated for 10 min, and the absorbance was quantified at 412 nm using a UV–Vis spectrometer. GSH concentration was determined using a standard curve of GSH (Sigma-Aldrich, St. Louis, MO, USA). GSH content in each sample was normalized with the total protein content and expressed in µmol/mg protein.

### 3.8. Cadmium Analysis

After ten days of Cd exposure, the hydroponic solution was removed and acidified to pH 2.0 with concentrated HNO_3_ (Sigma-Aldrich, St. Louis, MO, USA; purity: ≥99.999%). Then, the roots of the control and treated *T. latifolia* plants were washed with 0.1 M EDTA for 15 min, separated from the shoots, and both samples were dried at 70 °C for 24 h. Dried roots were digested with concentrated HNO_3_ and 30% H_2_O_2_, according to the methodology described by Rodríguez-Hernández, et al. [17]. Dried shoots were digested with HCl (Sigma-Aldrich, St. Louis, MO, USA; purity: ≥99.999%) and HNO_3_ mixture (3:1 *v*/*v*) according to Carranza-Álvarez, et al. [60]. All samples were analyzed by air-acetylene flame AAS (Varian SpectrAA 220FS, Mulgrave, Australia) at a wavelength of 228.8 nm. Cadmium concentration in the samples was estimated using a standard curve of Cd. Cd concentration was normalized with the dry weight in the plant samples and expressed in mg/Kg DW. Cd was also quantified in initial and residual media. All these data obtained were used to calculate the removal percentage (%), the translocation factor (FT), and the bioconcentration factor (BCF). Cd removal percentage was calculated according to the following formula:(4)$$\%\;\mathrm{Removal}=\frac{\left(\mathrm{Ci}-\mathrm{Cf}\right)}{\mathrm{Ci}}\times 100$$
where Ci is the Cd (mg/L) concentration in the initial medium, Cf is the concentration of residual Cd (mg/L) in the growth medium after treatment.

Translocation factor (TF) was calculated by dividing the concentration of total Cd (mg/Kg DW) in shoots by the concentration of total Cd (mg/Kg DW) in roots [31]. Bioconcentration factor (BCF) was determined by dividing the total Cd in roots (mg/Kg DW) by the total Cd present in the solution (mg/L) after treatment [12].

### 3.9. Bacteria Growth in the Hydroponic Systems and Root Colonization

The number of viable bacteria (CFU) in the hydroponic solution was estimated by serial dilutions according to Miles, et al. [61]. After Cd exposition, the number of bacteria associated with the roots of inoculated plants was determined according to Balsanelli, et al. [62]. The roots (150 mg) were harvested and homogenized aseptically in 1000 µL saline solution (0.85% NaCl). The extract was serially diluted and 10 µL of each dilution was plated on LB agar. The LB plates were incubated at 30 °C for 48 h.

### 3.10. Statistical Analysis

All experiments were performed in triplicate, and data were expressed as the mean ± standard deviation (SD). Statistical analyses were performed using the GraphPad Prism program (version 5.01, GraphPad Software Inc., San Diego, CA, USA). Statistical differences were determined by one-way ANOVA and Dunnett post hoc tests, or two-way ANOVA with Bonferroni post hoc tests.

## 4. Conclusions

In the present study, an axenic hydroponic culture of *T. latifolia* was established and used to evaluate the effect of *P. rhodesiae* GRC140 and exogenous IAA on plants exposed to Cd. Initially, we observed similar responses to those reported in *T. latifolia* exposed to Cd. We observed that Cd reduces shoot biomass in a dose-dependent manner and increases of electrolyte leakage and total GSH content in *T. latifolia* roots exposed at 20 and 40 mg/L of Cd.

On the other hand, *P. rhodesiae* GRC140 and exogenous IAA increase the removal and accumulation of Cd by *T. latifolia*. *P. rhodesiae* GRC140 increases the Cd content in the shoots, while the IAA increases the Cd content in the roots, similarly to that observed in other plant species inoculated with microorganisms and exposed to Cd. The axenic plants of *T. latifolia* in a hydroponic system could be used to evaluate the ability of the plant to remove other heavy metals and determine the effect of organic compounds in plant growth, and even establish assays of plant–microorganism interaction.

## Figures and Tables

**Figure 1 plants-11-01447-f001:**
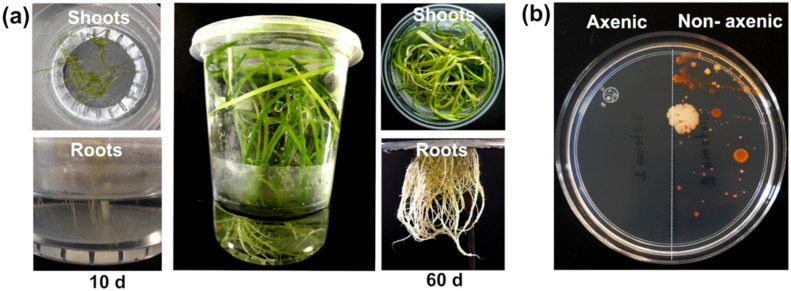
*T. latifolia* plants in an axenic hydroponic system (**a**) Side and top view of shoots and roots of ten and sixty-day-old *T. latifolia* plants in 0.2× MS medium pH 5.7 supplemented with 1.0% glucose. (**b**) Evaluation of axenic conditions of the nutrient solution after sixty cultivation days of *T. latifolia* plants.

**Figure 2 plants-11-01447-f002:**
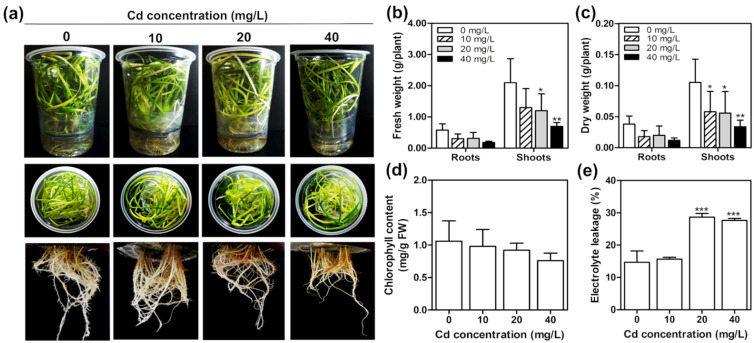
Cd effect on *T. latifolia* growth. Sixty-day-old *T. latifolia* plants were exposed to 10, 20, and 40 mg/L of Cd for ten days. (**a**) Shoot and root phenotype appearance, (**b**) fresh weight, (**c**) dry weight, (**d**) leaf chlorophyll content, (**e**) root electrolyte leakage of Cd exposed plants. The values represent the mean ± SD (*n* = 3). Asterisks indicate significant differences between control (0 mg/L) and treatment (Dunnett’s test; * *p* < 0.05, ** *p* < 0.01 and *** *p* < 0.001).

**Figure 3 plants-11-01447-f003:**
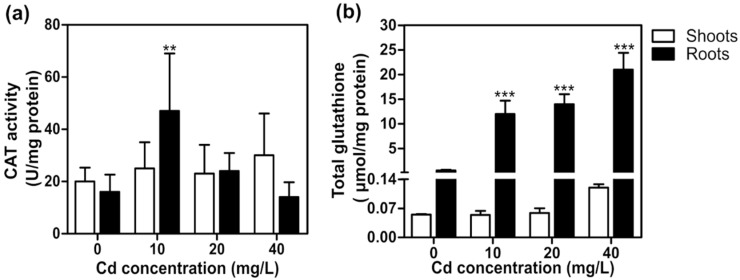
*T. latifolia* biochemical responses to Cd. Sixty-day-old *T. latifolia* plants were exposed to 10, 20, and 40 mg/L of Cd for ten days. (**a**) Catalase (CAT) activity and (**b**) total glutathione levels in shoots and roots. The values represent the mean ± SD (*n* = 3). Asterisks indicate significant differences between control (0 mg/L) and treatment (Dunnett’s test; ** *p* < 0.01 and *** *p* < 0.001).

**Figure 4 plants-11-01447-f004:**
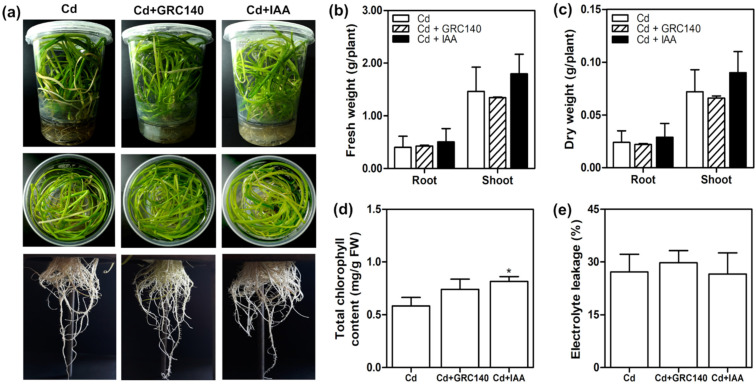
Effect of *P. rhodesiae* GRC140 and IAA on *T. latifolia* exposed to Cd. Sixty-day-old *T. latifolia* plants were treated with IAA (1.0 μM) or inoculated with *P. rhodesiae* GRC140 (10^6^ CFU/mL) for 6 days and exposed to 40 mg/L of Cd for ten days. (**a**) Shoot and root phenotype appearance, (**b**) fresh weight, (**c**) dry weight, (**d**) total chlorophyll content, (**e**) root electrolyte leakage. The values represent the mean ± SD (*n* = 3). Asterisks indicate significant differences between control (Cd) and treatment (Dunnett’s test; * *p* < 0.05).

**Figure 5 plants-11-01447-f005:**
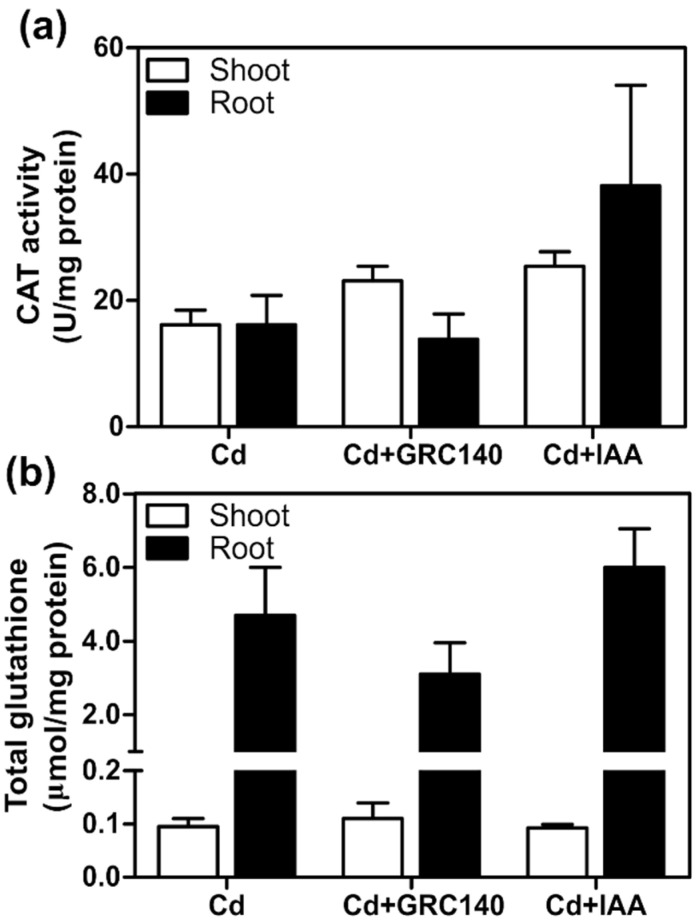
Effect of *P. rhodesiae* GRC140 and IAA on total glutathione and CAT activity in *T. latifolia* plants exposed to 40 mg/L of Cd for ten days. The values represent the mean ± SD (*n* = 3). Asterisks indicate significant differences between control (Cd) and treatment (Dunnett’s test). (**a**) CAT activity; (**b**) total glutathione.

**Figure 6 plants-11-01447-f006:**
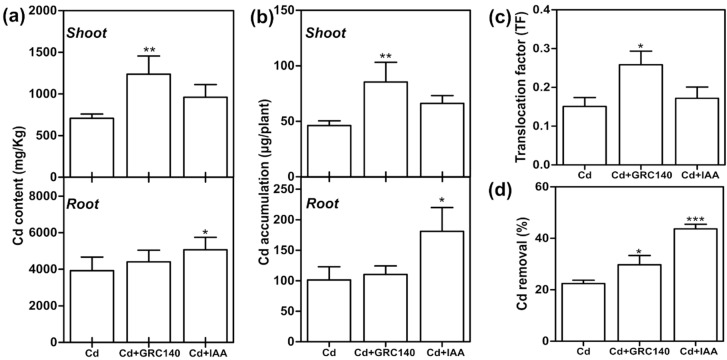
Effect of *P. rhodesiae* GRC140 and IAA on Cd content and Cd accumulation of *T. latifolia*. The values represent the mean ± SD (*n* = 3). Asterisks indicate significant differences between control (Cd) and treatment (Dunnett’s test; * *p* < 0.05, ** *p* < 0.01 and *** *p* < 0.001). (**a**) Cd content, (**b**) Cd accumulation, (**c**) translocation factor, (**d**) Cd removal.

**Table 1 plants-11-01447-t001:** Cd speciation in 0.2× MS medium at pH 5.7 predicted by Visual MINTEQ 3.1.

Species (%)	Cd Concentration (mg/L)		
	10	20	40
Cd^2+^	66.70	75.86	79.70
CdCl^+^	5.60	7.11	9.03
CdCl_2_ (aq)	0.03	0.04	0.06
CdSO_4_ (aq)	1.90	2.17	2.21
CdNH_3_^2+^	0.03	0.03	0.03
CdNO_3_^+^	1.06	1.20	1.25
CdHPO_4_ (aq)	1.42	1.60	1.64
CdEDTA^2−^	23.14	11.94	6.06
CdHEDTA^−^	0.07	0.04	0.02
Dissolved (%)	100	100	100
Precipitated (%)	0	0	0

aq: aquo complex.

**Table 2 plants-11-01447-t002:** Cd content in shoots and roots of *T. latifolia* plants exposed to Cd for 10 days.

Cd Concentration (mg/L)	Cd Content (mg/Kg DW)		BCF	TF	Cd Removal (%)
	Shoots	Roots			
0	nd	nd	nd	nd	nd
10	406 ± 59 ^b^	2970 ± 65 ^c^	509 ± 96 ^a^	0.18 ± 0.04 ^a^	37 ± 3.8 ^a^
20	668 ± 73 ^a^	3275 ± 15 ^b^	340 ± 38 ^b^	0.24 ± 0.01 ^a^	43 ± 6.3 ^a^
40	739 ± 24 ^a^	3520 ± 105 ^a^	126 ± 16 ^c^	0.23 ± 0.03 ^a^	22 ± 5.3 ^b^

BCF: bioconcentration factor; TF: translocation factor; nd: not detected. Values represent the mean ± SD (*n* = 3). Different letters represent statistically different means (*p* < 0.05, Tukey’s test).

## Data Availability

The data that support the findings of this study are available from the corresponding author, AHM, upon reasonable request.

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
