# Peer review of "Enhanced Cd-Accumulation in Typha latifolia by Interaction with Pseudomonas rhodesiae GRC140 under Axenic Hydroponic Conditions"

_plants, 2022, doi:10.3390/plants11111447_

Round 1

Reviewer 1 Report

Review report

Comments and suggestions to the authors

A brief summary

The environmental pollution and more specifically with heavy metals is among the most discussed issues nowadays. They are nondegradable and many plants, used from humans as basic food source, like rise from example, have the ability to accumulate them in their tissues.  Growing industrialization is considered as one of the main reasons for accumulation of toxic metals in the soil and water and therefore become a serious threat for human health.  Hence, the studies of different methods for reducing the concentration of toxic elements in the environment are essential. Because of this, my opinion is that the manuscript titled “Enhanced Cd-accumulation in Typha latifolia by interaction with Pseudomonas rhodesiae GRC140 under axenic hydroponic conditions” that was given to me for review is significant and very up-to-date.

The manuscript emphasizes on the role of phytoremediation in reducing the amount of Cd in the external substrate (hydroponic culture). The axenic hydroponic culture of Typha latifolia has been developed and cultured in media containing different concentrations of Cd. The obtained results proved the ability of this plant to reduce the Cd concentration with up to 43% in the external substrate and to accumulate it mainly in roots. Besides, the role of auxin-producing bacterium Pseudomonas rhodesiae GRC140 and exogenous IAA in Cd tolerance, translocation factor and bioconcentration factor in T. latifolia was also studied. The obtained results unequivocally prove that this bacterium could play a key role in increasing the plant’s ability to accumulate Cd and thus to reduce its concentration in the exogenous substrate.

My opinion is that studies reporting new and effective strategies in combating environmental pollutions are very necessary and should be supported. However, I have some minor remarks and suggestions:

Introduction:

Lane 39: “Therefore, different methods have been developed to reduce Cd concentrations in the environment and thus avoid its deleterious effects on human health and ecosystems” – please, give some concrete examples.

Lane 62:ACC – deaminase activity” – please, give the full name of this abbreviation.

Materials and Methods:

  • Section 3.2. “Then, seeds were germinated on agar culture flasks containing Murashige and Skoog (MS; Sigma-Aldrich, USA) salts at 0.2X strength, 3.5 mM 4-296 Morpholineethanesulfonic acid (MES; pH 5.7, Sigma-Aldrich, USA), 1.0 % (w/v) glucose, 297 and 0.8 % (w/v) Phytagar (Sigma-Aldrich, USA).” – please, specify what is the volume of the agar media.
  • Section 3.4.

Lane 308: please, clarify why 6-days incubation in the medium without glucose is necessary?

  • Section 3.5.

Lane 321: what is the origin of Pseudomonas rhodesiae GRC140?

Lane 321: the reference should be given with number.

Lane 326: what is the number of the transferred plants? I suppose 3, but it should be mentioned.

Lane 328: “The hydroponic systems were inoculated with 1.5 mL of 0.01 M MgSO4 solution (control), 1.5 mL of P. rhodesiae GRC140 suspension in 0.01 M MgSO4 or supplemented with indole-3-acetic acid (IAA) at a final concentration of 0.1 μM. “ – may be here is more appropriate to use “and” instead of “or”.

  • Section 3.6.

Lane 339: Fresh leaves (100 mg) were homogenized….” - In which samples the chlorophyll content has been measured? You should note here that total chlorophyll content has been measured in all controls and treated plants.

  • Section 3.7.

Lane 346: the same remark as above. Please, specify the roots from which plants?

  • Section 3.8.

Lane 359: the same remark as above. Plant tissues from which samples?

  • Section 3.9.

Lane 401: “the total Cd present in the solution (mg/L) after treatment” - Please, clarify is this Cf value? The Cd residues in the growth media?

  • Section 3.10

Lane 406: the root from which plants? Please, specify here;

Lane 407: I suggest here to use the term “homogenized” instead of “crushed”;

Line 407: 1 mL (1000 uL) is too small volume for initial suspension. Please, give information what quantity of roots (in mg/mkg) you have suspended in this volume (1000 uL).

Line 408: “The LB plates were incubated at 30 °C for 48 h.” - Did you prove the species identification of the counted bacteria? LB is not a selective media and many bacterial species can grow on it without any significant differences. If your hydroponic systems have been contaminated during cultivation how will you differentiate the Pseudomonas colonies from the colonies of eventual undesired microflora?

Results and discussion

This section represents in logical order all obtained results. The discussion is appropriate and the references are well selected. The figures and tables are clear and understandable with minor exceptions, indicated below. However, I have a few remarks and suggestions:

  • Section 2.6.

Line 208-209: “At 16 days post-inoculation (dpi), the number of bacteria in the hydroponic medium was 6.6 ± 0.2 log CFU, while in T. latifolia roots was 3.9 ± 0.02 log CFU of P. rhodesiae GRC140 (Figure 4).” - I recommend the number of bacteria to be given with lg instead log CFU. The decimal logarithm is more correct and understandable. For example: if the number of bacteria is 109 CFU/mL you can present it as lg 9 or log10 9, but not only log CFU. Besides, this is according to recommendations of ISO 80000.

Line 209: Figure 4 is not informative enough. I suggest to remake the figure and to present it as diagram where the difference between bacterial number in medium and in the roots will be clearer.

Line 219 – 223: “After Cd exposition, T. latifolia plants inoculated with P. rhodesiae GRC140 (Cd + GRC140 treatment) showed a phenotype similar to plants of Cd + IAA and Cd treatments (Figure 5a), and did not detect significant changes in fresh weight, dry weight, total chlorophyll content and electrolyte leakage (Figure 5a-e). While exogenous IAA significantly increases only the total chlorophyll content in T. latifolia plants exposed to Cd (Figure 5d). “- Here I detect discrepancy. You wrote that the content of total chlorophyll in Cd+IAA system was significantly increased but I can't see such significant difference in Fig 5d. Besides, in the first sentence you wrote the following "did not detect significant changes in fresh weight, dry weight, total chlorophyll content and electrolyte leakage". Well, which information is the correct one? Please, clarify here!

Line 229: there is a technical mistake in Figure 5a – it should be Cd+IAA, not AIA.

  • Section 2.7.

Line 249: Please, give the full name of EAA.

The remarks and recommendations I wrote above do not aim to underestimate the value of the paper. In my opinion these minor adjustments and additions will increase the value of the article. As I wrote above that kind of studies should be supported.

Author Response

Dear anonymous Reviewer 1, please see the attachment file that included your suggestions in the improved version of the manuscript. The changes are highlighted in yellow.

Reviewer 2 Report

Dear Authors,

General comments

The manuscript entitled "Enhanced Cd-accumulation in Typha latifolia by interaction with Pseudomonas rhodesiae GRC140 under axenic hydroponic conditions" presents interesting data on how to improve the ability of Typha latifolia plants to remove potentially toxic cadmium ions from the growth environment. Plants were grown under strictly controlled laboratory conditions - in hydroponic axemic system. The authors investigated the effects of three concentrations of cadmium salt, a precursor to various ionic forms of this element. In all cases, this element was taken up from the environment with high efficiency and accumulated by these plants (mainly in the roots). Low concentrations (10, 20 mg/L), as opposed to high concentration (40) of cadmium salts, usually did not have a negative effect on the tested physiological and biochemical parameters of these plants.

Infection with P. rhodesiae GRC140 bacteria enhanced the ability of these plants to take up cadmium ions and improved translocation of this element to the stem. The presented research has not only a cognitive but also a practical aspect. Despite comparable biological effects, under natural conditions, inoculating a plant with bacteria is simpler and more effective than using its metabolite, i.e. the IAA hormone. The manuscript is clearly written and the results are presented in the correct form. In my opinion, it can be published with minor additions and corrections in the Platnts journal.

Detailed comments

Line 66 - write this sentence in a different way

The authors explained what could be the cause of the decrease in catalase activity when plants were grown in the presence of high concentrations of cadmium, but did not refer to the increase in the activity of this enzyme at low concentrations of this toxic element. Please explain, referring to the relevant literature data, what may be the reason for such a large increase in the activity of this enzyme ??? Is catalase an enzyme induced under heavy metal stress in plants?

Line 174 - the authors studied the accumulation not in two tissues but in two organs: the root and the stem

Table 2. Under the table, explain the meaning of the abbreviations: BCF and TF

Line 189 - complete the sentence, add the metal of the given values

Explain the meaning of abbreviations first used in the manuscript text

Line 227 in this context the word: correlation cannot be used because the authors did not calculate the correlation coefficients between the examined features

In the methodology, provide the formula for calculating the content of chlorophyll a and b

line 377 - use reaction mixture instead of the reaction

Author Response

Dear anonymous Reviewer 2, please see the attachment file that included your suggestions in the improved version of the manuscript. The changes are highlighted in yellow.

Reviewer 3 Report

This study assesses the phytoremediation capacity of Cd by T latifolia species in combination with Pseudomonas rhodesiae GRC140 and IAA. The strong points of the article are represented by the clear and concise manner of presenting the results, both in the text and in the figures and tables. It is an interesting study that brings added value to the field of phytoremediation of Cd by elucidating the physiological and biochemical mechanisms underlying this process. It is a very well written study and I would like to congratulate the authors. I have only one specific comment, to add the title / description of the figure from the supplementary material.

Author Response

Dear anonymous Reviewer 3, please see the attachment file that included your suggestions in the improved version of the manuscript. The changes are highlighted in yellow.
